# SARS-CoV-2 Viral RNA Is Detected in the Bone Marrow in Post-Mortem Samples Using RT-LAMP

**DOI:** 10.3390/diagnostics12020515

**Published:** 2022-02-17

**Authors:** Tomasz Jurek, Marta Rorat, Łukasz Szleszkowski, Miron Tokarski, Izabela Pielka, Małgorzata Małodobra-Mazur

**Affiliations:** 1Department of Forensic Medicine, Wroclaw Medical University, 50-345 Wrocław, Poland; tomasz.jurek@umw.edu.pl (T.J.); marta.rorat@umw.edu.pl (M.R.); lukasz.szleszkowski@umw.edu.pl (Ł.S.); 2Genomtec Inc., Bierutowska Street 57-59, 51-317 Wroclaw, Poland; m.tokarski@genomtec.com (M.T.); i.pielka@genomtec.com (I.P.); 3Department of Molecular Techniques, Wroclaw Medical University, 50-367 Wrocław, Poland

**Keywords:** SARS-CoV-2, bone marrow, RT-LAMP, detection, postmortem samples

## Abstract

Since the emergence of severe acute respiratory syndrome coronavirus 2 (SARS-CoV-2) in late 2019, viral RNA has been detected in several different human tissues and organs. This study reports the detection of SARS-CoV-2 RNA in the bone marrow. Post-mortem samples were taken in a sterile manner during two forensic autopsies from the nasopharyngeal region, vitreous humor, cerebrospinal fluid, and bone marrow. SARS-CoV-2 was subsequently diagnosed via Genomtec^®^ SARS-CoV-2 EvaGreen^®^ RT-LAMP CE-IVD Duo Kit. In both postmortem patients, SARS-CoV-2 RNA was detected in bone marrow samples. However, both the vitreous humor and cerebrospinal fluid from the same patients gave negative results using the same test system. The evidence of viral RNA in the bone marrow, along with other reports supports the thesis that SARS-CoV-2 infections are systemic in nature, the consequences of which would profoundly influence both the testing and survival of patients.

## 1. Introduction

COVID-19 disease, caused by SARS-CoV-2 is a highly contagious infectious disease with a relatively high rate of death [1]. It is considered not only as a respiratory tract disease that affects the lungs but as a systemic disease affecting multiple organs such as the heart, brain, nervous system, liver, kidney, digestion truck, and others [2]. Since the emergence of severe acute respiratory syndrome coronavirus 2 (SARS-CoV-2) in late 2019, viral RNA has been detected in several different human tissues and organs. Except for lungs, RNA of SARS-CoV-2 was detected in the liver, heart, kidney, spleen, pancreas, gallbladder, small and large intestine, common carotid artery, and others [2]. For some specific tissue locations, it is hard to obtain biological samples from living subjects, that is why numerous reports describe SARS-CoV-2 detection postmortem, which allows determining almost all possible organs in terms of the presents of SARS-CoV-2 and possible consequences of COVID-19 [2].

The nucleic acid amplification tests (NAAT) utilizing nasopharyngeal swabs are the most common and sensitive methods for confirming the SARS-CoV-2 infection [3], with the most widely used technique for SARS-CoV-2 diagnosis being Real-Time PCR. However, other amplification techniques have recently been introduced, characterized by higher specificity and sensitivity, with a shorter time to results. These methods include loop-mediated isothermal amplification (LAMP) [4]. 

Viral RNA detection can provide vital information on modes of transmission and more importantly highlight previously hidden pools of infection within the human body, thus improving the prognosis and survival rate of COVID-19 patients. In our study we search the SARS-CoV-2 presents in not typical organs that little has been done so far, that is vitreous humor, cerebrospinal fluid, and bone marrow. We used postmortem samples for our study.

## 2. Materials and Methods

The protocol of this study was approved by the Bioethical Committee Board of Wroclaw Medical University (No. KB826/2020). 

### 2.1. Sample Characterization

Post-mortem samples were taken in a sterile manner during two forensic autopsies. Nasopharyngeal swabs (N), vitreous humor (V), cerebrospinal fluid (C) by sub-occipital puncture, and bone marrow (B) by aspiration from the posterior superior iliac crest were collected. Samples were taken 12 days after death and immediately placed in transport media (UTM, Biocomma), then transferred to the laboratory, maintaining a chilled environment throughout. The medical records have also been analyzed.

### 2.2. RNA Extraction and Ampflication

Viral RNA was extracted using QIAamp^®^ DSP Virus Spin Kit (Qiagen, Hilden, Germany) according to the manufacturer’s protocol. Eluted RNA was subjected to nucleic acid amplification using Genomtec^®^ SARS-CoV-2 EvaGreen^®^ RT-LAMP CE-IVD Duo Kit (Genomtec, Wroclaw, Poland) according to manufacturer’s instruction. Two SARS-CoV-2-specific gene fragments (*N* gene and *S* gene) and one human *RPP30* gene (inhibition control) were amplified using RT-LAMP. The diagnostic process was performed using the CFX96 Dx in vitro diagnostic system (BioRad, Hercules, CA, USA). Dedicated software was used for results visualization and interpretation. Amplification results were confirmed by 2% agarose gel electrophoresis. 

## 3. Results

Case 1. An 89-year-old man with hypertension, senile dementia and gout died within 19 days of COVID-19 diagnosis (confirmed by RT-PCR). Laboratory diagnostics performed during hospitalization did not reveal significant abnormalities except a moderate increase in inflammatory markers and lymphopenia. The main post-mortem pathologic findings were: severe, dense, consolidating infiltrates in both lungs; diffused alveolar damages in each lobe and edema in the bronchial mucosa; cardiomegaly; splenomegaly; and liver steatosis. The cause of death was acute respiratory distress syndrome. When analyzed via LAMP amplification, SARS-CoV-2 RNA was detected in bone marrow (B), but not in the vitreous humor (V) or cerebrospinal fluid (C) (Figure 1A,C, Table 1).

Case 2. A 91-year-old woman with chronic circulatory failure, ischemic heart disease, chronic kidney disease, type 2 diabetes with asymptomatic SARS-CoV-2 infection, died 9 days after a positive RT-PCR nasopharyngeal test. There were no post-mortem findings indicative of COVID-19, although advanced atherosclerosis, chronic pulmonary emphysema, multinodular goiter, liver steatosis, and pancreatic fibrosis were diagnosed. The cause of death was diabetic ketoacidosis. When analyzed SARS-CoV-2 RNA was detected in bone marrow (B) and nasopharyngeal swab (N), but not in the vitreous humor (V) or cerebrospinal fluid (C) (Figure 1B,C, Table 1).

Summarizing, in both patients SARS-CoV-2 RNA was detected postmortem in the bone marrow and for patient 2 the virus was also found in the nasopharyngeal swab. However, in neither case was viral RNA detected in either vitreous humor or cerebrospinal fluid. The positive control of SARS-CoV-2 RNA (Quantitative PCR (qPCR) Control RNA from Heat-Inactivated SARS-Related Coronavirus 2, Isolate USA-WA1/2020) gave a positive signal, and the no-template controls (NTC) were negative for both tests.

## 4. Discussion

Despite both SARS-CoV-2 patients being diagnosed over one week before death and the samples being taken 12 days after death, SARS-CoV-2 RNA remained detectable in the bone marrow of both examined patients. Significant disparities in the clinical course did not affect the detection of the RNA. The presence of viral RNA in the bone marrow and nasopharyngeal swab (case 2 only) indicates that both might represent equally advantageous environments for the virus to survive and replicate. In other published data, the presence of viral RNA has been detected in a variety of clinical samples including the lungs, bronchi, lymph nodes, spleen, heart, liver, kidney, brain, feces, blood, urine, and semen [5,6,7,8,9]. All the current research indicates that SARS-CoV-2 may spread via the bloodstream throughout the organism. This systemic spread would allow for the infection of the bone marrow, which due to its immune-related functions, would likely have a significant impact on the type and intensity of inflammatory response, thus affecting the chances of survival.

Some reports indicated previously that SARS-CoV-2 might be detected in the bone marrow. Deinhardt-Emmer et al. [10] reported SARS-CoV-2 in bone marrow in three out of eleven investigated samples collected postmortem. On the other hand, Massoth et al. [11] did not confirm the presence of SARS-CoV-2 in bone marrow collected postmortem either using RNA In situ hybridization or immunohistochemistry techniques. However, the negative results might be also related to the low sensitivity used in the study methods, as in both techniques there is no amplification of viral RNA.

In this preliminary communication, we used the isothermal method that allowed for a semi-quantitative assessment of viral RNA, with the positive test obtained in both cases with a time to result of below 30 minutes from the start of the amplification process. Further conclusions will be possible after future in-depth research including the testing of a larger population of the infected patients, quantitative assessment of genetic material, viral culture, detecting virus components, assessment of infection and replication, and potentially the characterization of inflammatory changes in the bone marrow.

## 5. Conclusions

The evidence of viral RNA in the bone marrow found during this study confirms that SARS-CoV-2 infections may indeed be systemic in nature, and thus the detection of the virus in this location could have a significant influence on future strategies not only in viral biology research but also in therapeutic interventions.

## Figures and Tables

**Figure 1 diagnostics-12-00515-f001:**
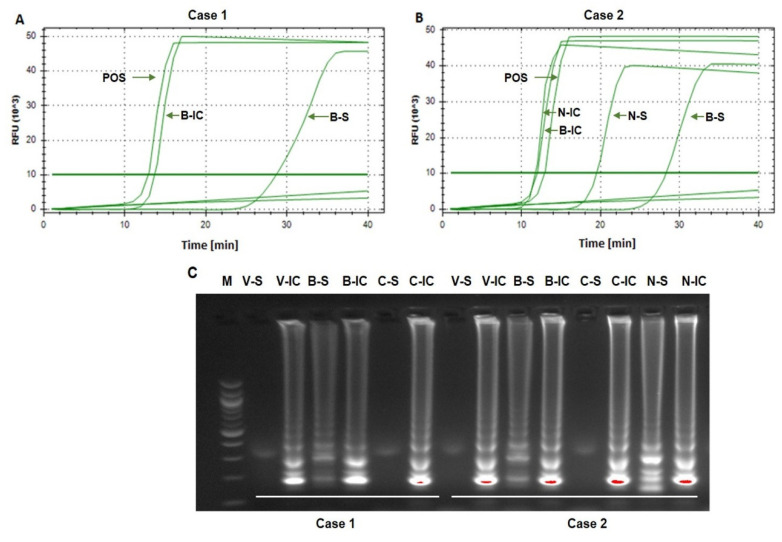
SARS-CoV-2 detection in postmortem collected samples. The real-time SARS-CoV-2 detection using Genomtec^®^ SARS-CoV-2 EvaGreen^®^ RT-LAMP CE-IVD Duo Kit from two clinical cases ((**A**)—case 1, (**B**)—case 2). The confirmation of obtained results by agarose gel electrophoresis (**C**). POS–positive sample with the standard genetic material of SARS-CoV-2; B-IC—bone marrow—inhibition control; B-S—bone marrow–SARS-CoV-2; N-IC—nasopharyngeal swab–inhibition control; N-S–nasopharyngeal swab—SARS-CoV-2; V-IC—vitreous humor—inhibition control; V-S—vitreous humor—SARS-CoV-2; C-IC–Cerebrospinal fluid—inhibition control; C-S—Cerebrospinal fluid—SARS-CoV-2.

**Table 1 diagnostics-12-00515-t001:** The Real-Time LAMP results of analyzed clinical cases. The table represents the time to detection of a particular sample using Genomtec^®^ SARS-CoV-2 EvaGreen^®^ RT-LAMP CE-IVD Duo Kit (N/A–not detected).

Sample	Case 1 Time to Results [min]		Case 2 Time to Results [min]	
	SARS-CoV-2	IC	SARS-CoV-2	IC
Nasopharyngeal swab (N)	Not tested	Not tested	19.48	11.91
Vitreous humor (V)	N/A	22.02	N/A	17.21
Cerebrospinal fluid (C)	N/A	17.18	N/A	14.33
Bone marrow (B)	28.70	13.67	28.28	11.73
Positive Control *	12.45			
NTC	N/A			

* (Quantitative PCR (qPCR) Control RNA from Heat-Inactivated SARS-Related Coronavirus 2, Isolate USA-WA1/2020, BEI Resources); NTC—No-template Control; IC—internal control; N/A—Not detected.

## Data Availability

Data are available on request from the corresponding author.

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
