# Peer review of "SARS-CoV-2 Viral RNA Is Detected in the Bone Marrow in Post-Mortem Samples Using RT-LAMP"

_diagnostics, 2022, doi:10.3390/diagnostics12020515_

Round 1

Reviewer 1 Report

Manuscript ID: diagnostics 1594350           

Detection of SARS-CoV-2 in the bone marrow in post-2 mortem samples using RT-LAMP

General comments: The case report of Jurek et al., discusses data on SARS-CoV-2 RNA detection from several post-mortem human tissues and organs (nasopharyngeal region, vitreous humor, cerebrospinal fluid and bone marrow) by RT-LAMP. The authors demonstrated the presence of viral RNA in the bone marrow, supposing that the SARS-CoV-2 infections in this site might have a significant impact on COVID-19 prognosis; because it could influence the type and intensity of inflammatory response in SARS-CoV-2 positive patients. Despite the role of SARS-CoV-2 infection in the bone marrow needs to be confirmed in a larger study cohort by molecular and virological approaches; the data reported are interesting and require further investigation. The case report is well presented and written. In my opinion, minor aspects of the manuscript need to be improved.

Results: The authors report in the results section that in both case reports (1 and 2) was not detected viral RNA in vitreous humor and cerebrospinal fluid samples. Have the authors confirmed this result with other molecular tests? Also, different from others clinical specimens, these samples could be need to be concentrated before molecular detection because viral RNA might be highly diluted. The authors used ultracentrifugation approch to confirm the complete absence of viral genetic material in the two negative samples by RT-LAMP?

Author Response

On behalf of all authors, thank you very much for your thorough review of the manuscript.

Thank you very much for appreciating the advantages of the presented for the evaluation manuscript and for all of the suggestions that will help for its improvement.

Answers to reviewer

General comments: The case report of Jurek et al., discusses data on SARS-CoV-2 RNA detection from several post-mortem human tissues and organs (nasopharyngeal region, vitreous humor, cerebrospinal fluid and bone marrow) by RT-LAMP. The authors demonstrated the presence of viral RNA in the bone marrow, supposing that the SARS-CoV-2 infections in this site might have a significant impact on COVID-19 prognosis; because it could influence the type and intensity of inflammatory response in SARS-CoV-2 positive patients. Despite the role of SARS-CoV-2 infection in the bone marrow needs to be confirmed in a larger study cohort by molecular and virological approaches; the data reported are interesting and require further investigation. The case report is well presented and written. In my opinion, minor aspects of the manuscript need to be improved.

Thank you for the general comments. Yes, we know that confirmation of the presence of SARS CoV-2 RNA in bone marrow needs to be done on a larger group, and, in the best-case scenario, correlate it with the severity of covid 19 disease or the prognosis of covid-19.

Results: The authors report in the results section that in both case reports (1 and 2) was not detected viral RNA in vitreous humor and cerebrospinal fluid samples. Have the authors confirmed this result with other molecular tests? Also, different from others clinical specimens, these samples could be need to be concentrated before molecular detection because viral RNA might be highly diluted. The authors used ultracentrifugation approch to confirm the complete absence of viral genetic material in the two negative samples by RT-LAMP?

We assumed that the RT-LAMP assay used for the study was characterized by the low limit of detection (LOD) for that reason we did not confirm the results with another test. This is a good idea of ultracentrifugation of the sample and we will use this approach for the next analysis.

Reviewer 2 Report

In the manuscript titled “Detection of SARS-CoV-2 in the bone marrow in post-mortem samples using RT-LAMP”, the author demonstrated the detection of SARS-CoV-2 viral RNA in several tissues form post-mortem samples. I feel it would be accepted after minor revisions, and the following is mistake which should be corrected:

  1. The title is misleading, as SARS-CoV-2 viral RNA in several human tissues and organs has been detected, but only the bone marrow is mentioned in title. In fact, bone marrow samples remained consistent and positive for SARS-CoV-2 viral RNA in both patients among several samples from different sources. So it is better to revise the title to “SARS-CoV-2 viral RNA is detected in the bone marrow in post-mortem samples using RT-LAMP”.

Author Response

On behalf of all authors, thank you very much for your thorough review of the manuscript.

Thank you very much for appreciating the advantages of the presented for the evaluation manuscript and for all of the suggestions that will help for its improvement.

Answers to reviewer

  1. The title is misleading, as SARS-CoV-2 viral RNA in several human tissues and organs has been detected, but only the bone marrow is mentioned in title. In fact, bone marrow samples remained consistent and positive for SARS-CoV-2 viral RNA in both patients among several samples from different sources. So it is better to revise the title to “SARS-CoV-2 viral RNA is detected in the bone marrow in post-mortem samples using RT-LAMP”.

Thank you for your suggestion, we have changed the title according to your suggestion.

Reviewer 3 Report

In this case report Jurek at al., detected SARS-CoV-2 RNA in different biological specimens, i.e. nasopharyngeal tract, vitreous humor, cerebrospinal fluid and bone marrow, from two dead bodies. The molecular test used for SARS-CoV-2 detection was RT-LAMP. This item of study, such as bone marrow,  must be of interest to investigate the pathogenesis role of systematic viral infection. However, I report some minor and major limitations for the pubblication of this case report.

Line 46: "improving the chances of detection and patient survival". As the same authors report in the introduction, lower and upper respiratory tract specimens are defined as biological materials of reference in the diagnosis of SARS-CoV-2 because the first viral localization is the lung. I recommend to reframe this sentence.

Line 46-47: it is missing the reference. I suggest this: Perchetti GA et al. "Validation of SARS-CoV-2 detection across multiple specimen types" to add at yours 5-8.

Line 48-49: vitreous humor and cerebrospinal fluid are investigated similarly in the study that I report above.

Line 58: I suggest to report the transport media with more details.

Line 62: The kit mentioned is intended for use of viral nucleic acids extraction from plasma or serum in vitro diagnostic use. If the same kit was used with satisfactory performance for different biological samples, as vitreous humor, cerebrospinal fluid and bone marrow, please you to include the reference.

Line 71-94: on the basis of these results, there are two cases with confirmed ante-mortem SARS-CoV-2 RNA by means of RT-PCR, but with documented cause of death for COVID-19 only for one case which it is negative by mean of RT-LAMP on nasopharyngeal swab. It would have been useful to confirm this negativity with molecular test on lower respiratory tract samples, i.e. BAL or BAS. Are there any lower respiratory sample to perform the same tests on both study cases? I report these interesting studies: 

-Middleton P, Perez-Guzman PN, Cheng A, Kumar N, Kont MD, Daunt A, et al. Characteristics and outcomes of clinically diagnosed RT-PCR swab negative COVID-19: a retrospective cohort study. Sci Rep. 2021; doi:10.1038/s41598-021-81930-0.

-Hase R, Kurita T, Muranaka E, Sasazawa H, Mito H, Yano Y. A case of imported COVID-19 diagnosed by PCR-positive lower respiratory specimen but with PCR-negative throat swabs. Infect Dis (Lond). 2020; doi:10.1080/23744235.2020.1744711.

Table 1: time of positivity for both bone marrow samples is very long comparing it to the positive control. As Jurek at al. affirm in the line 129-133, it would be interesting to quantify these positive samples and to compare with the ante-mortem last positive nasopharyngeal samples, is it possible? Finally, of very interest, it would be to test their viral replication and cytopathic effect on cell cultures, but this additional experiment should be difficult and long to perform.

Line 115-118: I approve this sentence and it would be of very interest a study only on the response of human bone marrow cells to SARS-CoV-2 RNA infection/positivity.

Line 135-138: the presence of SARS-CoV-2 RNA in the bone marrow samples not proves but confirms the systemic localization of the virus after the several studies that they documented its presence in the blood. 

Author Response

Manuscript ID: diagnostics 1594350

Detection of SARS-CoV-2 in the bone marrow in post- mortem samples using RT-LAMP

On behalf of all authors, thank you very much for your thorough review of the manuscript.

Thank you very much for appreciating the advantages of the presented for the evaluation manuscript and for all of the suggestions that will help for its improvement.

Answers to reviewer

Line 46: "improving the chances of detection and patient survival". As the same authors report in the introduction, lower and upper respiratory tract specimens are defined as biological materials of reference in the diagnosis of SARS-CoV-2 because the first viral localization is the lung. I recommend to reframe this sentence.

Thank you for your comments. The sentence has been changed. 

Line 46-47: it is missing the reference. I suggest this: Perchetti GA et al. "Validation of SARS-CoV-2 detection across multiple specimen types" to add at yours 5-8.

The mentioned publication has been cited in suggested position and added to the references list.

Line 48-49: vitreous humor and cerebrospinal fluid are investigated similarly in the study that I report above.

Line 58: I suggest to report the transport media with more details.

The transport media was specified.

Line 62: The kit mentioned is intended for use of viral nucleic acids extraction from plasma or serum in vitro diagnostic use. If the same kit was used with satisfactory performance for different biological samples, as vitreous humor, cerebrospinal fluid and bone marrow, please you to include the reference.

Please specify what reference you had in mind?

Line 71-94: on the basis of these results, there are two cases with confirmed ante-mortem SARS-CoV-2 RNA by means of RT-PCR, but with documented cause of death for COVID-19 only for one case which it is negative by mean of RT-LAMP on nasopharyngeal swab. It would have been useful to confirm this negativity with molecular test on lower respiratory tract samples, i.e. BAL or BAS. Are there any lower respiratory sample to perform the same tests on both study cases? I report these interesting studies: 

This is a very good point, but unfortunately, we don’t have this type of samples. We will take it into account in the future.

Table 1: time of positivity for both bone marrow samples is very long comparing it to the positive control. As Jurek at al. affirm in the line 129-133, it would be interesting to quantify these positive samples and to compare with the ante-mortem last positive nasopharyngeal samples, is it possible? Finally, of very interest, it would be to test their viral replication and cytopathic effect on cell cultures, but this additional experiment should be difficult and long to perform.

 The longer time of bone marrow samples is related to the lower copy number or genome equivalent of the SARS-CoV-2. It is quite difficult to quantify as more diagnostic kits allow for quality analysis, not quantity. We are not the BSL3 laboratory dedicated to SARS-CoV-2 culturing. We don’t have the ante-mortem samples as well, that is why we are not able to perform proposed experiments, but I agree that this is a very interesting approach and it would increase our knowledge about SARS-CoV-2 infection.     

Line 115-118: I approve this sentence and it would be of very interest a study only on the response of human bone marrow cells to SARS-CoV-2 RNA infection/positivity.

Thank you for approving. Good point of study.

Line 135-138: the presence of SARS-CoV-2 RNA in the bone marrow samples not proves but confirms the systemic localization of the virus after several studies that documented its presence in the blood. 

It was corrected according to the reviewer suggestion

Round 2

Reviewer 3 Report

Previous Reviewer Comment

Line 46-47: it is missing the reference. I suggest this: Perchetti GA et al. "Validation of SARS-CoV-2 detection across multiple specimen types" to add at yours 5-8.

Author’s response

The mentioned publication has been cited in suggested position and added to the references list.

2 Reviewer’s response

The reference “Perchetti G.A., Nalla K.A., Huang M.L., Zhu H., Wei Y., Stensland L., Loprieno M.A., Jerome K.R., Greninger A.L. Validation 181 of SARS-CoV-2 detection across multiple specimen types. J Clin Virol, 2020, 128, 104438, doi: 10.1016/j.jcv.2020.104438” is suggested for the sentence “To our best knowledge, this study is the first to report the detection of SARS-CoV-2 RNA in the bone marrow.” that it is deleted.

Previous Reviewer Comment

Line 62: The kit mentioned is intended for use of viral nucleic acids extraction from plasma or serum in vitro diagnostic use. If the same kit was used with satisfactory performance for different biological samples, as vitreous humor, cerebrospinal fluid and bone marrow, please you to include the reference.

Author’s response

Please specify what reference you had in mind?

2 Reviewer’s response

It is your competence to find the correct reference for the procedure used.

Previous Reviewer Comment

Table 1: time of positivity for both bone marrow samples is very long comparing it to the positive control. As Jurek at al. affirm in the line 129-133, it would be interesting to quantify these positive samples and to compare with the ante-mortem last positive nasopharyngeal samples, is it possible? Finally, of very interest, it would be to test their viral replication and cytopathic effect on cell cultures, but this additional experiment should be difficult and long to perform.

Author’s response

The longer time of bone marrow samples is related to the lower copy number or genome equivalent of the SARS-CoV-2. It is quite difficult to quantify as more diagnostic kits allow for quality analysis, not quantity. We are not the BSL3 laboratory dedicated to SARS-CoV-2 culturing. We don’t have the ante-mortem samples as well, that is why we are not able to perform proposed experiments, but I agree that this is a very interesting approach and it would increase our knowledge about SARS-CoV-2 infection.  

2 Reviewer’s response

I would like to highlight that the quantification of the SARS-CoV-2 can be performed in the LAB of SARS-CoV-2 diagnosis, BSL-3 safety level, without viral cell culture using molecular kits, i.e. Clonit quanty-19v2 Detection an

Author Response

Manuscript ID: diagnostics 1594350

Detection of SARS-CoV-2 in the bone marrow in post- mortem samples using RT-LAMP

On behalf of all authors, thank you very much for your thorough review of the manuscript.

Thank you very much for appreciating the advantages of the presented for the evaluation manuscript and for all of the suggestions that will help for its improvement.

Answers to reviewer

Line 46-47: it is missing the reference. I suggest this: Perchetti GA et al. "Validation of SARS-CoV-2 detection across multiple specimen types" to add at yours 5-8.

Author’s response

The mentioned publication has been cited in suggested position and added to the references list.

2 Reviewer’s response

The reference “Perchetti G.A., Nalla K.A., Huang M.L., Zhu H., Wei Y., Stensland L., Loprieno M.A., Jerome K.R., Greninger A.L. Validation 181 of SARS-CoV-2 detection across multiple specimen types. J Clin Virol, 2020, 128, 104438, doi: 10.1016/j.jcv.2020.104438” is suggested for the sentence “To our best knowledge, this study is the first to report the detection of SARS-CoV-2 RNA in the bone marrow.” that it is deleted.

At the discussion stage, the authors realized that this is not the first report describing detection of SARS-CoV RNA in bone marrow, for that reason, this sentensce was removed. According to the authors, suggested by the reviewer citation also match its current place.  

Previous Reviewer Comment

Line 62: The kit mentioned is intended for use of viral nucleic acids extraction from plasma or serum in vitro diagnostic use. If the same kit was used with satisfactory performance for different biological samples, as vitreous humor, cerebrospinal fluid and bone marrow, please you to include the reference.

Author’s response

Please specify what reference you had in mind?

2 Reviewer’s response

It is your competence to find the correct reference for the procedure used.

There is a limited number of RNA extraction kits dedicated for such unusual samples, especially taken post-mortem dedicated for IVD, for that reason we were not able to match the appropriate kit to the appropriate type of biological material. As the control, we used the internal (inhibition control) as the sign of successful RNA extraction, which was positive for all analyzed samples.

Previous Reviewer Comment

Table 1: time of positivity for both bone marrow samples is very long comparing it to the positive control. As Jurek at al. affirm in the line 129-133, it would be interesting to quantify these positive samples and to compare with the ante-mortem last positive nasopharyngeal samples, is it possible? Finally, of very interest, it would be to test their viral replication and cytopathic effect on cell cultures, but this additional experiment should be difficult and long to perform.

Author’s response

The longer time of bone marrow samples is related to the lower copy number or genome equivalent of the SARS-CoV-2. It is quite difficult to quantify as more diagnostic kits allow for quality analysis, not quantity. We are not the BSL3 laboratory dedicated to SARS-CoV-2 culturing. We don’t have the ante-mortem samples as well, that is why we are not able to perform proposed experiments, but I agree that this is a very interesting approach and it would increase our knowledge about SARS-CoV-2 infection.  

2 Reviewer’s response

I would like to highlight that the quantification of the SARS-CoV-2 can be performed in the LAB of SARS-CoV-2 diagnosis, BSL-3 safety level, without viral cell culture using molecular kits, i.e. Clonit quanty-19v2 Detection an

Thank you for this suggestion, we will for sure include it in our next study. Unfortunately at the moment, we are not able to quantify the RNA copies number in analyzed samples.